# The Advance of Single-Cell RNA Sequencing Applications in Ocular Physiology and Disease Research

**DOI:** 10.3390/biom15081120

**Published:** 2025-08-04

**Authors:** Ying Cheng, Sihan Gu, Xueqing Lu, Cheng Pei

**Affiliations:** Department of Ophthalmology, The First Affiliated Hospital of Xi’an Jiaotong University, Xi’an 710061, China; cying514@xjtufh.edu.cn (Y.C.); gusihan@stu.xjtu.edu.cn (S.G.); lxq1523@stu.xjtu.edu.cn (X.L.)

**Keywords:** single-cell RNA sequencing, cellular heterogeneity, ocular physiology, ocular disease, biomarker

## Abstract

The eye, a complex organ essential for visual perception, is composed of diverse cell populations with specialized functions; however, the complex interplay between these cellular components and their underlying molecular mechanisms remains largely elusive. Traditional biotechnologies, such as bulk RNA sequencing and in vitro models, are limited in capturing cellular heterogeneity or accurately mimicking the complexity of human ophthalmic diseases. The advent of single-cell RNA sequencing (scRNA-seq) has revolutionized ocular research by enabling high-resolution analysis at the single-cell level, uncovering cellular heterogeneity, and identifying disease-specific gene profiles. In this review, we provide a review of scRNA-seq application advancement in ocular physiology and pathology, highlighting its role in elucidating the molecular mechanisms of various ocular diseases, including myopia, ocular surface and corneal diseases, glaucoma, uveitis, retinal diseases, and ocular tumors. By providing novel insights into cellular diversity, gene expression dynamics, and cell–cell interactions, scRNA-seq has facilitated the identification of novel biomarkers and therapeutic targets, and the further integration of scRNA-seq with other omics technologies holds promise for deepening our understanding of ocular health and diseases.

## 1. Introduction

The eye comprises a variety of functionally distinct cell populations, such as the corneal cell group that maintains transparency and is involved in refraction, the aqueous humor outflow cell group that regulates intraocular pressure by controlling aqueous humor drainage, the photoreceptor cell group including cone and rod cells responsible for central and peripheral vision as well as color and low-light perception, etc. These various cell populations within the eye are intricately organized and functionally specialized, working together through a complex network of interactions to form a sophisticated visual information processing system [1,2]. However, the molecular features and regulatory networks of different cell subclusters and their interaction are still largely unknown. The study of cellular heterogeneity is relatively challenging due to the requirement of analytical approaches at the single-cell level, for example, traditional bulk RNA sequencing techniques cannot distinguish gene expression profiles of individual cells in vivo, limiting the identification of specific disease-related cell types and their typical biomarkers, as well as the verification of the dynamic alternation of cellular states [3]. In vitro cell culture models, while providing a foundation for molecular research, could not fully mimic the complexity of human ophthalmic diseases, limiting a deep understanding of disease mechanisms and the development of new therapies.

In recent years, the development of scRNA-seq technology has provided unprecedented opportunities for ophthalmic research [1]. The advantage of scRNA-seq lies in its ability to provide high resolution at a single-cell level, revealing cellular heterogeneity, capturing the gene expression characteristics of individual cells, and thus playing an important role in the identification of cell type specificity, disease-related genes, and molecular pathways [4]. With the growing application of scRNA-seq in ophthalmology, this review aims to provide an overview of scRNA-seq technology application in ophthalmological research, thus offering an in-depth understanding of the underlying cell/molecular mechanisms and potential therapeutic targets.

## 2. scRNA-Seq Technology

By isolating individual cells and sequencing their RNA, scRNA-seq unveils the transcriptomic heterogeneity within cellular populations. This approach offers distinct advantages over traditional bulk RNA sequencing [5]. The scRNA-seq workflow starts with preparing a single-cell suspension from ocular tissue samples, and enzymes such as papain are commonly used to dissociate cells gently to ensure minimal damage (Figure 1). Following the isolation and lysis of individual cells, cellular RNA is reverse-transcribed using primers incorporating unique cell barcodes. In Drop-Seq, developed by Macosko et al. [6], individual cells are co-encapsulated with barcoded microparticles in nanoliter droplets; primers conjugated to microparticles have a 12 bp cell barcode and an 8 bp unique molecular identifier (UMI), which distinguishes biological transcripts from Polymerase Chain Reaction (PCR) artifacts. This combinatorial barcoding strategy enables the unambiguous assignment of each sequenced transcript to its original single cell. Building on this core barcoding principle, commercial platforms like 10× Genomics Chromium and BD Rhapsody^TM^ now dominate the field, offering robust, standardized solutions. While both leverage microfluidics and barcoded beads for high-throughput capture, the two methods employ distinct strategies: 10 × Chromium typically uses oil droplets for partitioning with fixed barcode assignment, whereas BD Rhapsody often relies on microwell arrays, allowing for random cell-bead pairing and integrated imaging capabilities. The barcoded cDNA is then amplified to enhance sequencing sensitivity. High-throughput sequencing generates short-read sequences, which are then processed using bioinformatics tools. These tools enable analytical workflows that begin by aligning sequences to a reference genome to generate gene expression matrices. Quality control is then applied. Subsequent dimensionality reduction, using techniques like PCA or t-SNE, facilitates the visualization of sample clusters. Finally, differential expression analysis identifies genes with significant functional relevance. This comprehensive analysis not only uncovers the structural organization and inherent heterogeneity of cellular populations but also pinpoints critical gene expression variations between distinct cell types [4,5,7,8].

Complementing scRNA-seq, single-nucleus RNA sequencing (snRNA-seq) isolates nuclei rather than whole cells [9]. This approach enables the transcriptome profiling of frozen/archived specimens and tissues resistant to gentle dissociation, particularly valuable for delicate ocular samples like the neuronal-rich retina. Although primarily capturing nuclear transcripts, snRNA-seq effectively resolves cellular heterogeneity when intact cell isolation is impractical, making it a vital tool alongside scRNA-seq. In ophthalmology, scRNA-seq and its expanding snRNA-seq hold great potential, providing detailed insights into cellular and molecular mechanisms of ocular development, physiological homeostasis, and pathological alterations. In this review, we underscore the latest advances of scRNA-seq application in ophthalmologic research (Figure 2), improving our understanding of underlying mechanisms in ocular physiology and disease states, and the potential role of scRNA-seq in the exploration of new therapeutic strategies.

## 3. Ocular Tissue Development

The coordinated interactions of various cell types and precise spatiotemporal regulation during ocular tissue development are crucial for the establishment and maintenance of visual function. In this section, we review the recent applications of scRNA-seq in ocular tissue development, with a primary focus on the cornea, the retina, and the lens, particularly emphasizing retinal development and the photoreceptors within the retina.

Limbal progenitor cells (LPCs) are vital for maintaining corneal epithelial homeostasis and facilitating wound healing, and scRNA-seq has revealed their cellular heterogeneity and dynamic differentiation hierarchy, unmasking stage-specific gene expression programs that govern their self-renewal and commitment to corneal epithelial lineages. Li et al. [10] revealed the molecular trajectories of LPC differentiation and identified *Retinoic Acid-Related Orphan Receptor Alpha (RORA)* as a key molecular switch that drives LPCs to differentiate into mature corneal epithelial cells (CECs) by activating *PITX1* and establishing specific enhancers and chromatin interactions. Their work deepened our understanding of transcriptional dynamics and 3D epigenetic regulation mediated by *RORA* during differentiation. Meanwhile, Collins et al. [11] also used scRNA-seq but combined it with the assay for transposase-accessible chromatin sequencing (ATAC-seq) to generate a single-cell atlas of the human cornea spanning from embryonic to adult stages. This study identified new markers for LPCs, such as *GPHA2*, a key surface marker for maintaining the undifferentiated state of LPCs, which is crucial for epithelial regeneration. In addition, scRNA-seq also helps analyze human iPSC-derived corneal organoids during their maturation process and confirms similar features of organoids with the normal cornea, thus serving as preliminary evidence supporting the potential application of organoids in corneal transplantation [12].

The application of scRNA-seq in lens development is still in its preliminary stage. Lens epithelial cells (LECs) are located on the surface of the anterior capsule of the lens, and possess the ability to proliferate and differentiate into lens fiber cells (LFCs), which is crucial for maintaining the transparency and function of the lens. Zhu et al. [13] utilized scRNA-seq to uncover two distinct differentiation trajectories of LECs into LFCs and identified a subpopulation of *ADAMTSL4*^+^ cells with stem/progenitor cell characteristics, marked by the expression of genes such as *MKI67* and *STMN1*, offering novel insights into lens development. Moreover, by manipulating the gene expression patterns of *ADAMTSL4*^+^ cells and other key LEC subsets, it could be feasible to restore the normal lens development process in cataract patients, thus providing a new approach for lens regeneration therapy and potentially reducing the reliance on traditional cataract surgeries.

The wide application of scRNA-seq is progressively uncovering the mechanisms behind various aspects of retinal development. Zhang et al. [14] revealed that *Jarid2* drives the transition of retinal progenitors from early to late stages by mediating H3K27me3 deposition and repressing early genes like *Foxp1*. Another study utilizing *Vglut1*^−/−^ and *Gnat1*^−/−^ retinal organoids showed that glutamatergic neuronal activity regulates retinal angiogenesis and blood-retinal barrier maturation via Norrin/b-catenin signaling [15]. As for the retinal ganglion cell (RGC) differentiation, a critical transcription factor *ATOH7* has long been identified as early as 2001 [16], and scRNA-seq further enriched our understanding of this process. Deletion of the *ATOH7* enhancer leads to RGC deficiency and optic nerve hypoplasia in mice by using ATAC-seq and scRNA-seq [17]; a subsequent study further confirmed its role in the human retina and its organoids, indicating that *ATOH7* is also involved in the specification of late-born cones in addition to its known role in RGCs [18]. These findings underscore the intertwined roles of epigenetic modifications and neurotransmitter signaling in retinal development, aligning with the mainstream view that epigenetic landscapes and neurochemical cues act as hierarchical regulators of cellular fate and tissue patterning. Several studies on photoreceptor development highlight *NR2E3* and *IGF1* as critical regulators, and *NR2E3* ensures the proper expression of phototransduction genes [19], while *IGF1* promotes photoreceptor specification via *PHLDA1*, which inhibits AKT phosphorylation to guide retinal progenitor cells into cone/rod fates [20]. These findings collectively reveal a hierarchical regulatory network in photoreceptor development, where *NR2E3*-driven transcriptional programming and *IGF1*-mediated post-translational control intersect to enforce both the spatial-temporal precision of progenitor cell fate decisions and the functional maturation of light-sensing machinery. The integration of nuclear receptor signaling by *NR2E3* with cytoplasmic kinase cascades through *IGF1*/*AKT* highlights how multi-layered regulatory mechanisms, such as spanning transcriptional, epigenetic, and post-translational levels, collaboratively orchestrate cellular specialization.

Recent advances in multi-omics technologies, including spatial transcriptomics, chromatin accessibility assays, and multiplexed protein localization techniques, have enabled a more comprehensive and high-resolution characterization of cellular heterogeneity and molecular mechanisms during retinal development. Dorgau et al. [21,22] integrated scRNA-seq with spatial transcriptomics to map human retinal organoid development, identifying a putative ciliary margin as the niche for transient retinal progenitor cells (RPCs) expressing *ZIC1* and *HES6*. These early RPCs give rise to diverse retinal neurons under the regulation of *SOX/PRDM/TEAD* transcription factors. Additionally, multiplexed protein mapping techniques were also used combined with scRNA-seq to show that the *OTX2* regulon is a key regulator of RGC fate by promoting photoreceptor and bipolar cell programs while inhibiting RGC, amacrine cell, and horizontal cell fates [23]. These integrated multi-omics approaches not only dissect the spatiotemporal regulatory logic of retinal neurogenesis, but also establish a high-resolution molecular profile for modeling human retinal development.

## 4. Application of scRNA-Seq in Ocular Diseases

The burgeoning literature of scRNA-seq has been reported in ophthalmologic disease research. Here, we will make a detailed description and discussion of this cutting-edge technology and its latest advances, facilitating a more nuanced understanding of cell and molecular heterogeneity in various ocular diseases. In addition, some representative scRNA-seq studies with general analysis approaches are summarized in Table 1 for a better understanding.

### 4.1. Myopia

The key mechanisms contributing to myopia development underlined the complex interplay of genetic and environmental factors, and the typical pathological features of myopia development were sclera remodeling, induced by fibroblast-to-myofibroblast differentiation and reduced collagen production. *CCDC66*, which has been identified to be related to retinal atrophy and degeneration, emerges as a candidate gene that may contribute to high myopia [24,25]. By using scRNA-seq, the results showed that *CCDC66* was consistently expressed in the embryonic retinas of mice, and its mutations disrupt cell proliferation by affecting the mitotic process, which provides a plausible mechanism for high myopia onset [26]. Furthermore, the newly identified *PI16*^+^/*SFRP4*^+^ fibroblast subpopulation is genetically involved in extreme myopia, a special type defined as an SE of less than or equal to −10.00 D, or AL of more than or equal to 28.00 mm [27]. This finding underscores the heterogeneity within fibroblast populations contributing to pathological scleral changes. Moreover, scRNA-seq helps identify a high myopia-specific *Apoe*^+^ rod subpopulation and the disrupted ON/OFF signaling driven by cone-bipolar cells from highly myopic mouse models; the cell–cell communication network revealed attenuated neuronal interactions and increased glial-vascular interactions in highly myopic retinas [28].

Lens abnormalities, such as lens shape, thickness, or refractive index changes, can contribute to refractive errors by altering the eye’s total refractive power. A previous study used a total of 50,375 single cells from the lens epithelium of highly myopic and control mouse eyes, revealing a significant increase in the proportion of lens fiber cells and the top down-regulated gene *Notch2* of lens epithelium in highly myopic eyes [29]. Another study conducted the electroacupuncture treatment in lens-induced myopia guinea pigs—this approach could significantly attenuate myopia progression by improving retinal mitochondrial function, such as the upregulation of *COX3*, *ND4*, and *ND2*, etc. [30]. The above studies strengthen the evidence of the lens-related factors in myopia development, with structural and molecular alterations, potentially disrupting optical homeostasis. Future studies should explore whether lens-specific molecular changes directly initiate or merely exacerbate refractive errors, and whether combining lens-focused and neuroprotective therapies could yield synergistic effects.

### 4.2. Ocular Surface and Corneal Diseases

The recent application of scRNA-seq has unveiled the intricate cellular heterogeneity and molecular landscapes within the cornea and its limbal region, which is essential for ocular surface integrity and immune surveillance. Ligocki et al. [31] identified 16 distinct cell clusters that include stromal keratocytes, endothelium, and various epithelial subtypes in adult human cornea, offering a nuanced view of corneal cellular characteristics. A unique entity of transit-amplifying cells (TACs) in the corneal limbus was also characterized through scRNA-seq, revealing a set of cell cycle-dependent genes that serve as TAC signature markers, and these markers are pivotal for tissue homeostasis and regeneration [32]. Further expanding the immunological perspective, the diverse immune cell subsets within the corneal limbus indicated a complex immunological microenvironment essential for defending against infections and maintaining ocular surface health [33].

Several common ocular surface disorders were explicitly studied as well, including pterygium, dry eye disease (DED), Fuchs’ endothelial corneal dystrophy (FECD), etc. In a recent study, a total of 45,605 cells from both pterygium patients and conjunctival controls were analyzed, revealing nine major cellular types and their subtypes, underscoring the increased presence of immune cells and the activation of specific cellular subtypes such as immuno-fibroblasts, epithelial–mesenchymal transition-related epithelial cells, and activated vascular endothelial cells in pterygium. Notably, the interaction between macrophages and *ACKR1*^+^-activated endothelial cells was identified as a key player in pterygium development [34]. These findings enriched the pterygium’s paradigm from a degenerative disorder to a dynamic cellular ecosystem with potential druggable interventions.

DED is a prevalent chronic eye disease characterized by an aberrant inflammatory response in the ocular surface. The presence of macrophages and fibroblast-like epithelial cell subtypes with pro-inflammatory characteristics during both acute and chronic phases in conjunctival tissues from the DED mice model has been identified [35]. Furthermore, Liu et al. [35] revealed the dominant CD4^+^ T-cell clusters, including T helper 1 (Th1), Th17, and regulatory T (Treg), and three distinct macrophage subtypes, with the CD72^+^CD11c^+^ subtype contributing to chronic inflammation. Keratoconus, traditionally considered a non-inflammatory corneal thinning disorder, is increasingly recognized to involve inflammatory processes in its pathogenesis. The immune features of keratoconus have also been explored by both scRNA-seq and bulk RNA-seq analyses. A total of 10 immune-associated genes, such as *CEP112*, *FYN*, *IFITM1*, *IGFBP5*, etc., were identified to relate to the prognosis; these genes were related to cytokine and MMP profile expression, indicating vital participation in the keratoconus progression [36]. The effective management of FECD is a challenging job due to its genetic heterogeneity and different progression rates. scRNA-seq applications uncover the specific subtypes with associated gene expression features, cell heterogeneity, molecular mechanisms, stages and severities, and largely assist the ophthalmologist in the decision-making and clinical management of patients with FECD.

### 4.3. Glaucoma

The ocular outflow tissues, such as the trabecular meshwork (TM) and Schlemm’s canal (SC), are important components responsible for regulating intraocular pressure (IOP) by facilitating the drainage of aqueous humor. However, it is impractical to dissect individual tissue types in the aqueous humor outflow structure due to their small size and complex anatomical structure, hence, most previous studies of the aqueous humor outflow pathway at the genetic level mainly employed bulk-RNA sequencing or in vitro cell culture, such as TM cells [37,38,39].

Recent advancements in scRNA-seq offer novel insights into the cellular heterogeneity and molecular characteristics of ocular outflow tissues. Patel et al. [40] generated a cell atlas of human conventional outflow tissues from 17,757 genes across 8758 cells from eight human donors’ eyes by using scRNA-seq, and they identified 12 distinct cell types and mapped the region-specific expression of candidate genes. In addition, there are two different expression patterns, including myofibroblast- and fibroblast-like cells in TM, and SC exhibited a unique feature with both lymphatic and blood vascular gene expressions. Similarly, the presence of predominant lymphatic-endothelial features of SC cells in C57BL/6J mice was also identified [41]. In addition, Jia et al. [42] verified the tissue contractility and dysfunction of TM in glaucomatous non-human primates. This dual phenotype enables TM-SC to efficiently regulate aqueous humor outflow and maintain intraocular pressure, with lymphatic-like properties contributing to specialized functions such as valve formation and fluid drainage. Moreover, it has been proposed that TM-SC may possess contractile capabilities, biomechanical sensing or paracrine signaling like vascular endothelia, rather than merely serving as a static conduit. Dysregulation of lymphatic-endothelial features may accelerate the progression of glaucoma pathogenesis.

It is widely accepted that increased resistance to outflow occurs mainly in the TM and post-TM regions. Understanding the cellular and molecular mechanisms governing these pathways is critical for developing targeted glaucoma therapies that restore outflow facility and control IOP effectively. van Zyl et al. [43] used scRNA-seq to investigate the cellular composition of the aqueous humor outflow pathways and gene expression profile in human eyes and other four glaucomatous animal models—they verified the immunological milieu and the lymphatic characteristics of the conventional outflow pathway, and *Foxc2* gene mutations are primarily linked to lymphatic vasculature dysfunction, considered as a risk gene in congenital glaucoma [44,45]. The dual involvement of *Foxc2* in both neural crest-derived TM cells and SC endothelium implicated the intricate crosstalk in maintaining the drainage balance of aqueous humor [46]; in addition, ANGPT1-TEK signaling pathway involved in TM-SC crosstalk has also been confirmed to play a key regulatory role in IOP homeostasis [47], suggesting that defects in either TM or SC population can compromise outflow facility.

RGC loss and optic nerve cupping are the typical pathological alterations in glaucomatous progression. Recent studies prove that glaucoma shares similar immunological mechanisms with other neurodegenerative diseases, such as Alzheimer’s disease [48]. The retinal glial activation and infiltrating T-cell-mediated immune response play key roles in RGC damage in glaucoma. For instance, homeostatic microglia could shift toward an inflammatory phenotype under pathological conditions; their hyperreactivity and driven neuroinflammation are regarded as a strong contributor to RGC loss. With the assistance of scRNA-seq, Pan et al. [49] identified an important immune regulator, *IGFBPL1*, that plays a pivotal role as a master switch, converting inflammatory microglia to their homeostatic state to prevent excessive neuroinflammation in glaucoma; hence, *IGFBPL1* is a potential therapeutic target for neurodegenerative diseases featuring microglial dysregulation. Similarly, several promising regulatory molecules have been discovered. For example, *B3gat2* and *Tsc22d* genes were found to be associated with neuronal migration and inflammatory inhibition [50], and the knockdown of *Rock1* and *Rock2*, or aquaporin 1 and β2 adrenergic receptors, reduces the IOP and alleviates the RGC loss in glaucoma mice [51].

Collectively, scRNA-seq largely promotes the characteristics of TM and RGC heterogeneity in glaucoma, revealing disease-associated molecular changes. However, tissue dissociation may selectively lose critical, fragile neuronal subtypes. The combination with other omics technologies such as bulk RNA-seq, proteomics, and spatial transcriptomics enables a more comprehensive understanding of cell-level mechanisms underlying glaucomatous optic nerve damage.

### 4.4. Uveitis

Uveitis is a sight-threatening intraocular inflammatory disorder, featuring abnormalities in multiple immune cells infiltrating into the eye tissues such as aqueous humor, vitreous, choroid, and retina. Several important molecules and immune features in uveitis progression have been identified in the animal model using scRNA-seq, which is powerful in identifying infiltrating immune cell diversity, activation states, and specific biomarkers. Of note, the loss of fragile cells such as leukocytes, and the potential molecular feature alteration of immune cells during processing need to be paid attention to. Cervical draining lymph nodes (CDLNs) are the major lymph nodes of ocular drainage, which have been regarded as an important source of autoreactive lymphocytes during autoimmune diseases involving the eyes, especially uveitis. Based on the physiological principle, the biological samples commonly used for scRNA-seq analysis are aqueous humor and peripheral blood mononuclear cells in uveitis patients, and CDLNs and retinas in uveitis animal models such as the experimental autoimmune uveitis (EAU) model, which is a classic and widely used uveitis model induced by retinal antigen and adjuvants.

*Hypoxia-inducible factor 1 alpha (Hif1α)* is confirmed as an up-regulated EAU-associated differentially expressed gene (DEG) in Th1, Th17, and Treg cells, and enhanced *Hif1α* expression in CD4^+^ T cells was conserved and may promote their proliferation in uveitis patients [52]. Quinn et al. [53] performed scRNA-seq of retinas from EAU mice; the results highlighted that the significantly upregulated expressions of pro-inflammatory molecules, complement factors, and MHC class II were found in Müller glia, and retinal pigment epithelium (RPE) cells, and CellPhoneDB analysis revealed the close interactions between Müller cells and T cell subtypes, indicating their participation in interfering with inner and outer blood-retinal barriers and the potential breakdown of retinal immune privilege. Kang et al. [54] analyzed the aqueous humor of patients with uveitis associated with Behcet’s disease (BD) and Vogt–Koyanagi–Harada disease (VKHD), and they noticed that there are distinct landscapes of immune cell infiltration and T-cell clonal expansions between them, with VKHD showing a preference for pro-inflammatory CD4^+^ Th1 cells and BD for cytotoxic CD8^+^ T cells, confirming the existence of immunopathogenic discrepancy.

Currently, several therapeutic approaches applied in uveitis have been analyzed by scRNA-seq to evaluate their treatment effect and underlying mechanisms, including immunosuppressive agents, mesenchymal stem cells, recombinant proteins, etc. Cyclosporine A (CsA), mycophenolate mofetil (MMF), and dimethyl fumarate (DMF) are common immunosuppressive agents used in autoimmune diseases such as uveitis. scRNA-seq analysis unveiled that CsA partly reverses the changes in immune cell proportions including Th17 and Treg cells, and affects B cell differentiation and immunoglobulin secretion-its rescue impact on T cells is superior to glucocorticoids in EAU mice [55]; similar results were found in MMF or DMF-treated EAU mice, the downregulation of the EAU-associated molecules, a decrease in Th1 and Th17 cell proportions and inhibition of B cell function were presented in EAU mice after MMF treatment [56], and DMF treatment could partly reverse the effector T/Treg cell imbalance and inhibit the retina-infiltrating T cells [57]. Other novel treatment approaches, such as mesenchymal stem cells (MSCs), have also sparked new perspectives in uveitis therapy. Gao et al. [58] used gingiva-derived mesenchymal stem cells (GMSCs) to treat EAU mice via caudal vein injection, further scRNA-seq verified the immunosuppressive properties of GMSCs, which can rescue the Th17/Treg imbalance and suppress the pro-inflammatory Th17 subtype; a recent study showed that overexpressing a chemokine receptor, *CCR5*, in MSCs could enhance homing capacity and better attenuate EAU through their immunomodulatory capacity than regular MSCs, providing an innovative insight for MSC-based treatment for uveitis [59].

### 4.5. Retinal Diseases

The advent of scRNA-seq facilitates the plotting of the comprehensive cell atlases of the retina from human donors [60,61] and other mammals [62], providing a foundation for understanding the molecular underpinnings of retinal function and disease. Diabetic retinopathy (DR) is a serious complication of diabetes mellitus, characterized by the progressive dysfunction of retinal microvasculature, such as microaneurysm formation, macular edema, retinal hemorrhages, and neovascularization [63]. Translational models such as the Akimba mouse have provided valuable insights into DR pathogenesis, scRNA-seq analysis revealed cell-type-specific dysregulation in macroglial subpopulations, metabolic shifts from glycolysis to oxidative phosphorylation, and activation of oxidative stress and inflammatory pathways [64]. Recent scRNA-seq studies unveiled the presence of innate immune dysregulation in the peripheral blood of diabetic macular edema patients [65], and the underlying mechanism may be linked to the inflammatory response and abnormal microvasculature, and dysregulation of *CSF1*/*CSF1R* induced by glia and endothelial cells interaction [66]. Wang et al. [67] generated a retinal single-cell atlas to elucidate the cell–cell crosstalk of the inner blood-retinal barrier in the early stages of DR; they highlighted the importance of the *Ctxn3*^+^ Müller subgroup in DR pathology, and constructed the interaction net among endothelial cells, pericytes, and Müller subsets, clarifying the complex regulatory relationships between these cells. Corano et al. [68] used scRNA-seq to analyze the preretinal fibrovascular membranes surgically removed from patients with proliferative DR (PDR) stage, they identified the unique function of different cell subtypes, including the endothelial cells with an angiogenic profile, macrophages expressing proangiogenic cytokines, and a pericyte-myofibroblast transitioning subcluster regulated by *adipocyte enhancer-binding protein 1 (AEBP1),* highlighting the involvement of pericytes in fibrogenesis and the potential *AEBP1*-targeting therapeutic strategy in advanced diabetic retinopathy.

Age-related macular degeneration (AMD) primarily affects the macula, characterized by the loss of photoreceptors and RPE-choroid dysfunction. The etiology of AMD is multifactorial, involving genetic predispositions and environmental influences. To uncover the risk genes of AMD, integrative analyses of bulk and single-cell transcriptomics in the human eye were performed, along with the co-localization of genome-wide association studies (GWAS) and expression quantitative trait locus (eQTL), which identified 15 putative causal genes, of which *TSPAN10* and *TRPM1* were enriched in RPE and associated with the risk of AMD [69]. Another study obtained the human AMD single-cell transcriptomic datasets from the Gene Expression Omnibus (GEO) database and performed the secondary analysis; results revealed three potential novel cell markers including *DNASE1L3* for endothelial cells, *ABCB5* for melanocytes, and *SLC39A12* for RPE cells, and significant changes in cell abundance and crosstalk among fibroblasts, melanocytes and Schwann cells between AMD and control groups [70], in line with other studies that demonstrated the specific cell-type compositional and cell-specific gene expression changes in the progression of AMD [71,72]. Although the scRNA-seq enables cell-level dissection of RPE and photoreceptors, it lacks spatial resolution to fully capture macular region-specific pathology. Complementary approaches such as epigenomics, bulk-seq, and spatial omics could further enhance the molecular map of pathological features such as drusen and choroidal neovascularization.

**Table 1 biomolecules-15-01120-t001:** Summary of typical studies of ocular diseases using scRNA-seq.

Disease	Sample	Cell Number	Method	Key Findings	Reference
**Glaucoma**	Trabecular meshwork and neighboring tissues from 8 human eyes from 4 donors	8758	scRNA-seq	Schlemm’s canal exhibits a unique combination of lymphatic and vascular gene expression. Mapped key glaucoma-related genes to specific cell clusters, revealing their roles in IOP regulation.	Patel et al. [40] (2020)
	SECs from C57BL/6J and 129/Sj mouse strains	~4500 SECs (bulk RNA-seq); 903 single SECs (scRNA-seq and snRNA-seq)	bulk RNA-seq, scRNA-seq, snRNA-seq	Schlemm’s canal cells have a lymphatic-biased identity and identified key marker genes. Characterized interactions between SECs and trabecular meshwork cells, providing insights into the regulation of aqueous humor outflow and IOP.	Balasubramanian et al. [41] (2024)
	Anterior segment issues from 7 human eyes, cynomolgus macaque, rhesus macaque, pig, and mouse	24,023 (human); 5158 (rhesus macaque), 9155 (cynomolgus macaque), 6709 (pig), 5067 (mouse)	scRNA-seq	Generated a cell atlas of aqueous humor outflow pathways in humans and four model species. Exhibited conservation and differences in cell types and gene expression across species, providing insights into glaucoma pathogenesis.	Van et al. [43] (2020)
	Retinas from WT mice and *IGFBPL1* KO mice	3500 (scRNA-seq)	scRNA-seq combined with bulk RNA-seq	*IGFBPL1* deficiency leads to microglial activation and progressive neurodegeneration in the retina and brain, while *IGFBPL1* resets pro-inflammatory microglia to a homeostatic state via *IGF1R* signaling.	Pan et al. [49] (2023)
	Ciliary body and contiguous tissues from adult mice with ocular hypertension models	Not specified	scRNA-seq	CRISPR-CasRx-mediated disruption of *Aqp1*/*Adrb2*/*Rock1*/*Rock2* genes reduces IOP and RGC damage in mice.	Yao et al. [51] (2024)
**Uveitis**	Cervical draining lymph node cells from NC, EAU control, and EAU mice	47,048	scRNA-seq	*Hif1α* identified as a potential participant in autoimmune uveitis pathogenesis by regulating Th-17, Th1, and regulatory T cells.	Zhu et al. [52] (2023)
	Retina from 1 healthy mouse and 2 EAU mice for scRNA-seq; RPE from 3 healthy mice and 2 EAU mice for bulk RNA-seq	11,516 (scRNA-seq)	scRNA-seq combined with bulk RNA-seq	During EAU, interactions exist between Müller glia and T cell/natural killer cell subsets. RPE cells exhibited an epithelial-to-mesenchymal transition signature during EAU.	Quinn et al. [53] (2024)
	Aqueous humor from 3 BD patients and 3 VKHD patients	47,048	scRNA-seq combined with scTCR-seq	BD uveitis shows significant myeloid cell infiltration and CD8+ T cell clonality with cytotoxic phenotype, while VKHD uveitis is dominated by CD4+ T cells with Th1-like phenotype.	Kang et al. [54] (2023)
	3 healthy adult mice, 6 EAU model mice (3 untreated, 3 treated with CsA)	41,349	scRNA-seq	CsA reversed EAU-associated changes in immune cell composition and gene expression, and reduced the differentiation and immunoglobulin secretion of plasma B cells, and restored the balance between pathogenic T cells and Tregs.	Duan et al. [55] (2022)
	3 healthy adult mice, 6 EAU model mice (3 untreated, 3 treated with MMF)	41,349	scRNA-seq	MMF reduced the differentiation tendency from naïve to effector phenotypes and downregulated pathogenic cytokine production in Th1 and Th17 cells. MMF also inhibited B-cell immunoglobulin production and antigen processing and presentation.	Wang et al. [56] (2023)
	6 healthy adult mice, 12 EAU model mice (6 untreated, 6 treated with DMF)	41,349	scRNA-seq	DMF treatment effectively ameliorated EAU symptoms by reversing the Teff/Treg imbalance and inhibiting the ocular infiltration of Teff cells. DMF downregulated *PIM1* and *CXCR4* expression, which are critical for T cell differentiation and migration. DMF also reduced the proportion of plasma cells by inhibiting *PIM1* expression in B cells.	Zhu et al. [57] (2024)
	CDLNs from 9 normal mice, 9 EAU model mice (6 untreated, 3 treated with GMSCs)	71,000	scRNA-seq	GMSC treatment alleviated EAU symptoms, reduced retinal immune cell infiltration, and restored Th17/Treg balance by regulating Th17/Treg-related genes and suppressing pro-inflammatory Th17 cell formation.	Gao et al. [58] (2023)
	2 WT C57BL/6J mice and 2 EAU model mice	20,448	scRNA-seq	EAU immune response is primarily driven by Th1 cells, and *CCR5*-overexpressing MSCs enhance homing capacity and improve immunomodulatory outcomes in preventing EAU.	Yuan et al. [59] (2024)
**Retina** **diseases**	4 healthy human PBMCs, 4 DME human PBMCs	57,650	scRNA-seq	The presence of innate immune dysregulation in the peripheral blood of DME patients with T2D, and pro-inflammatory CD14+ monocytes predominated in promoting inflammation.	Ma et al. [65] (2021)
	10 retinas from 5 diabetic mice, 10 retinas from 5 WT mice	31,256	scRNA-seq	Identified two microglial subpopulations and three EC populations in retinal cells of diabetic retinopathy. Found *CSF1*/*CSF1R* crosstalk dysregulation associated with PDR.	Ben et al. [66] (2024)
	5 rat retinal samples (2 normal SD rats, 3 DR rats)	35,910	scRNA-seq	Constructed a communication network among ECs, pericytes, and two Müller cell subtypes in the early stage of DR.	Wang et al. [67] (2022)
	4 fibrovascular membranes from PDR patients	4044	scRNA-seq	Identified a subset of macrophages expressing proangiogenic cytokines and a pericyte-myofibroblast transdifferentiating subcluster.	Corano et al. [68] (2023)
	129 human donor eyes (106 control and 23 AMD patients) for bulk RNA-seq; 5 control donor eyes for snRNA-seq	100,055 nuclei (snRNA-seq)	snRNA-seq combined with bulk RNA-seq	Identified 15 putative causal genes for AMD, with some highly expressed in the RPE.	Orozco et al. [69] (2020)
	3 human donor eyes (2 controls and 1 AMD)	4335	scRNA-seq	Discovered three novel cell markers: *DNASE1L3* for ECs, *ABCB5* for melanocytes, and *SLC39A12* for RPE cells. Constructed cell-specific TF regulatory loops, highlighting key TFs in AMD pathogenesis.	Wang et al. [70] (2023)
	Retina from 2 adult humans for scRNA-seq, retina from 15 adult humans with and without AMD for bulk RNA-seq	92,385 (scRNA-seq)	scRNA-seq combined with bulk RNA-seq	Identified 9772 and 1214 differentially expressed genes in the macula and periphery for advanced AMD vs. control comparison. Enrichment in complement and coagulation, antigen presentation, tissue remodeling signaling pathways.	Lyu et al. [71] (2021)
	RPE/choroid from 3 human donors (2 normal eyes and 1 neovascular AMD eye) for scRNA-seq; RPE/choroid from 96 normal human donors for bulk RNA-seq	4766 (scRNA-seq)	scRNA-seq combined with bulk RNA-seq	Identified *VEGF*-, *BMP*-, and *tenascin*-mediated pathways as strong intercellular communication pathways related to aging and senescence. AMD samples showed higher senescence scores than normal cells	Dhirachaikulpanich et al. [72] (2022)
	Mouse retinas from the OIR model and control mice	76,164	scRNA-seq (BD Rhapsody platform)	Pericyte sub-population 2, highly expressing *Col1a1*, was vulnerable to pathological angiogenesis, and *Col1a1* expression was upregulated in the aqueous humor of patients with PDR or ROP.	Xia et al. [73] (2023)

Abbreviations: scRNA-seq, Single-cell RNA sequencing; IOP, Intraocular pressure; SECs, Schlemm’s canal endothelial cells; snRNA-seq, Single-nucleus RNA sequencing; WT, Wild type; IGFBPL1, Insulin-like growth factor binding protein-like 1; KO, knockout; RGC, Retinal ganglion cell; NC, Normal control; EAU, Experimental autoimmune uveoretinitis; Hif1α, Hypoxia-inducible factor 1 alpha; Th, T helper; RPE, Retinal pigment epithelium; BD, Behçet’s disease; VKHD, Vogt–Koyanagi–Harada disease; scTCR-seq, Single-cell T-cell receptor sequencing; CsA, Cyclosporine A; Treg, Regulatory T cells; MMF, Mycophenolate mofetil; DMF, Dimethyl fumarate; Teff, Effector T cells; CXCR4, C-X-C motif chemokine receptor 4; CDLNs, Cervical draining lymph nodes; GMSCs, Gingival mesenchymal stem cells; MSC, Mesenchymal stem cells; PBMCs, Peripheral blood mononuclear cells; DME, Diabetic macular edema; T2D, Type 2 diabetes; EC, Endothelial cell; CSF1, Colony-Stimulating Factor 1; CSF1R, Colony-Stimulating Factor 1 Receptor; PDR, Proliferative diabetic retinopathy; DR, Diabetic Retinopathy; AMD, Age-related macular degeneration; TF, Transcription Factor; VEGF, Vascular endothelial growth factor; BMP, Bone morphogenetic protein; OIR, Oxygen-induced retinopathy; ROP, Retinopathy of prematurity.

The oxygen-induced retinopathy (OIR) mouse model is commonly utilized for studying retinal diseases, particularly in retinopathy of prematurity (ROP) and PDR. The OIR model mimics the pathophysiological processes seen in human ischemic retinopathies and has become instrumental in understanding retinal angiogenesis. A study constructed a transcriptomic atlas from over 76,000 single cells across 4 mouse retinas, sub-clustering analysis revealed a novel pericyte subcluster closely susceptible to retinal capillary dysfunction with a marker gene of *Col1a1*. *Col1a1* silencing decreased the neovascular and avascular areas in OIR retinas and suppressed pericyte-myofibroblast transition and endothelial-mesenchymal transition in mice. Consistently, *Col1a1* is overexpressed in the aqueous humor of patients with PDR or ROP, suggesting a potential treatment target of retinal capillary dysfunction [73].

### 4.6. Ocular Tumor

Recent studies have utilized scRNA-seq to explore the cellular heterogeneity and immune microenvironments of ocular tumors, identifying key genes and cell subpopulations associated with tumor progression, metastasis, and prognosis. Conjunctival melanoma (CoM) is a rare and aggressive ocular malignancy that arises from melanocytes in the conjunctival epithelium. The current pathogenesis of CoM involves the activation of cancer-associated fibroblasts (CAFs) that bolster angiogenesis via VEGFR signaling, alongside an immunosuppressive microenvironment marked by decreased infiltration of functional CD8^+^ T cells. Shi et al. [74] used scRNA-seq to characterize the cellular landscape of CoM, highlighting increased CAFs in metastatic CoM, which linked to enhanced angiogenic capacity and increased VEGFR expression that drive tumor progression; additionally, they observed a relatively quiescent immunological environment with reduced total CD8^+^ T cells and increased naive CD8^+^ T cells, thus confirming the pathological and physiological characteristics of CoM at the single-cell level. Furthermore, another study analyzing 41 CoM samples and 11 normal nevus samples identified upregulated m6A demethylase fat mass and obesity-associated protein in CAFs, which promotes tumor neovascularization by erasing m6A modifications of pro-angiogenic factors such as *VEGFA* and *EGR1* [75].

Uveal melanoma (UM) is a highly aggressive intraocular malignancy originating from melanocytes in the uveal tract. The integration of scRNA-seq and machine learning provides a robust tool for elucidating the immune cell profile of UM. Chen et al. [76] utilized scRNA-seq to analyze 14 UM patient samples to elucidate the relationship between intratumoral CD8^+^ T cells and UM metastasis, revealing that the exhaustion of CD8^+^ T cells in the tumor microenvironment plays a crucial role in promoting metastasis, and developed a 3-gene model including *SLC25A38*, *EDNRB*, and *LURAP1* to predict metastatic risk and prognosis. Similarly, Li et al. [77] conducted a comprehensive analysis using scRNA-seq on 37,660 malignant cells from 17 UM tumor samples, uncovering distinct intra-tumoral subtypes with varying prognoses and immune microenvironments, and identified a 9-gene signature (including *BAP1*, *SF3B1*, etc.) that stratifies patients based on tumor cell heterogeneity, providing a valuable tool for personalized treatment strategies. Macrophages have a complex dual role in tumor progression. A scRNA-seq study [78] on macrophages in UM revealed four transcriptionally distinct macrophage subsets, with MΦ-C4 showing reduced expression of both M1 and M2 signature genes, indicating a loss of inflammatory and antigen-presenting functions, and instead demonstrating enhanced signaling for proliferation, mitochondrial functions, and metabolism, which correlated with aggressive tumor behavior and poor prognosis. Additionally, scRNA-seq has also offered prospects for targeted therapy. Li et al. [79] investigated the role of *BAP1* mutations in UM, and revealed that the *ITGB2*-*ICAM1* axis promotes liver metastasis by hypoxia and ECM regulation, suggesting that inhibiting this axis could offer a promising strategy for preventing metastasis in *BAP1*-mutated UM.

Retinoblastoma is the most common pediatric ocular malignancy, typically metastasizing through two primary routes, including direct invasion into contiguous structures and hematogenous dissemination to distant sites. Liu et al. [80] employed scRNA-seq to delineate the local extension mechanisms, highlighting chromosome 6p amplification and *SOX4* expression as key drivers. From the perspective of the immune microenvironment, Cuadrado et al. [81] confirmed the key immunosuppressive factors such as macrophage migration inhibitory factor and extracellular matrix metalloproteinase inducer in retinoblastoma. In terms of tumor metastasis, Yang et al. [82] identified cone precursors as the cell of origin and *UBE2C* as a pivotal factor in malignant transformation. These studies have unraveled the complex network of retinoblastoma metastasis from multiple dimensions, including molecular mechanisms, cellular origin, and immune microenvironment.

## 5. Summary and Prospect

We summarized the recent research findings concerning the application of scRNA-seq in ocular development and various ocular disease studies. This review offers a focused synthesis of the latest studies spanning the entire visual system, which integrates cutting-edge applications in pathomechanism research and therapeutic target identification to fill critical gaps in timeliness and translational relevance. The application of scRNA-seq in ocular research is promising due to its powerful analysis ability. However, scRNA-seq has certain limitations. First, the preparation of single-cell suspensions often requires exogenous digestive enzymes, which may alter cellular transcriptional profiles and obscure inherent disease-related changes. A further upgrade of the processing workflow is necessary. Second, it is susceptible to batch effects during sample processing, as variations in specimen collection time or sequencing runs can introduce technical noise. To reduce the batch effects, some approaches such as algorithms (e.g., Harmony [83], ComBat [84]) and emerging deep learning have been employed to improve data integration fidelity. Third, the technology lacks spatial context, failing to preserve the positional information of cells and thus hindering the interpretation of spatial heterogeneity and the specific roles of cells within tissue microenvironments. The integration of multi-omics is a promising way to complete the missing information.

As this technology continues to evolve, several promising prospects warrant attention. First, scRNA-seq will facilitate a deeper understanding of ocular development. Researchers can identify essential regulatory pathways and key genes that govern eye formation, contributing to breakthroughs in congenital eye disorders and developmental anomalies. Second, the application of scRNA-seq in studying ocular diseases is expected to expand, which could help identify novel biomarkers and therapeutic targets, thus paving the way for personalized medicine approaches. Next, integrating scRNA-seq with other omics technologies will largely advance our view of ocular health and diseases, such as single-cell assay for transposase accessible chromatin sequencing (scATAC-seq), 3D chromatin conformation mapping technology and base-resolution deep learning [85]. Spatial transcriptomics, genome, proteomics, and metabolomics can further join this integration [23]. Some public datasets, including EBI Single Cell Expression Atlas, CZ cellxgene, The Human Cell Atlas, Tabula sapiens, and Tabula muris, offer high-quality scRNA-seq data worldwide in a convenient way, thus saving the cost and accelerating the discoveries in the field. Last, this technology largely promotes the therapeutic development for ocular diseases in the aspects of novel gene target identification, personalized medicine exploration, in vitro disease model establishment, treatment effectiveness evaluation, and drug mechanisms and resistance research (Figure 3).

Altogether, scRNA-seq holds tremendous potential for advancing ocular research by providing detailed insights into cellular mechanisms, enhancing our understanding of disease processes, and facilitating the development of targeted therapies.

## Figures and Tables

**Figure 1 biomolecules-15-01120-f001:**
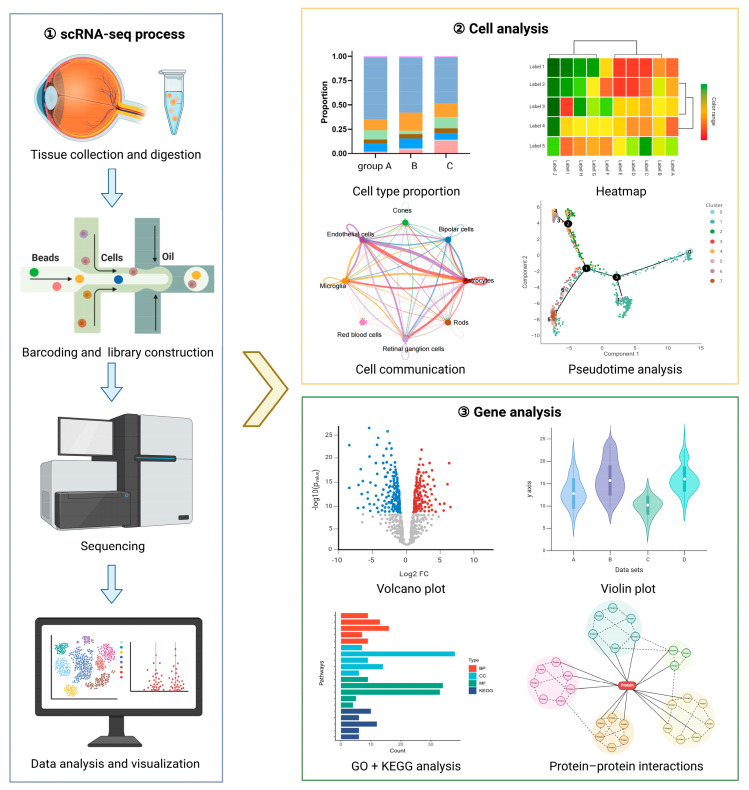
Schematic diagram of the scRNA-seq workflow. ① The specific eye tissue is collected and undergoes digestion to make the cell suspension. Each cell is labeled with a unique barcode during cDNA synthesis, and the cDNA library is constructed. Amplified libraries are submitted for scRNA-seq and the reads are aligned to a reference genome; the sequencing data are visualized and analyzed. ② Cell analysis is the main part of scRNA-seq study, commonly including cell subtype proportion, cell subtype difference, cell interaction, cell communication, pseudotime analysis, etc. ③ Gene analysis is a common and important part of scRNA-seq research, mainly including differentially expressed genes, functional enrichment analysis, protein–protein interactions, etc. Created in BioRender. Cheng, Y. (2025) https://BioRender.com/g23y304.

**Figure 2 biomolecules-15-01120-f002:**
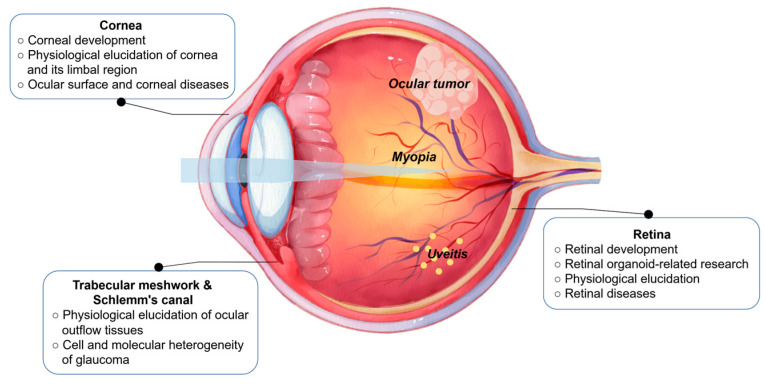
The main application of scRNA-seq in ocular research includes ocular tissue development, physiology, and ocular diseases. By Figdraw (https://www.figdraw.com/#/).

**Figure 3 biomolecules-15-01120-f003:**
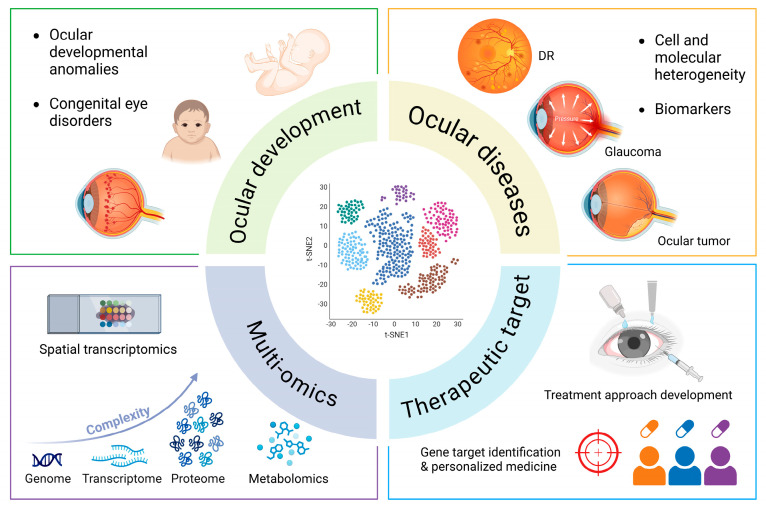
Important prospects of scRNA-seq application in ocular research include ocular development, ocular diseases, multi-omics integration, and therapeutic target exploration. Created in BioRender. Cheng, Y. (2025) https://BioRender.com/m79w206.

## Data Availability

No new data were created or analyzed in this study.

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
