# Peer review of "The Advance of Single-Cell RNA Sequencing Applications in Ocular Physiology and Disease Research"

_biomolecules, 2025, doi:10.3390/biom15081120_

Round 1
Reviewer 1 Report
Comments and Suggestions for Authors
As stated by the authors:
“The advent of single-cell RNA sequencing (scRNA-seq) has revolutionized ocular research by enabling high-resolution analysis at the single-cell level, uncovering cellular heterogeneity, and identifying disease-specific gene profiles”
- What about the need of the present review? Please specify the added value of the present review in respect to what already present in the literature
- Another important point when approaching a review is relative to the scientific experience of the authors in the field they are revising. Please, add information about each personal research contribution
- In general, the present review is is too long and vague spanning from eye development to ocular diseases including the anterior and the posterior part of the eye. The discussion should be more focused. A superficial survey does not bring much information to the reader.
- As an additional consideration, this long narrative review is difficult to revise in light of the application of a given technique such as the single-cell RNA sequencing
Reviewer 2 Report
Comments and Suggestions for Authors
scRNA-Seq has indeed transformed many areas of biomedical research and it is timely to review the many advances provided by this technology in vision science. It is noteworthy that the very first single cell RNA-Seq publication by Macosko et al involved the retina and I think it would be appropriate to cite this study (Macosko EZ, et al Highly Parallel Genome-wide Expression Profiling of Individual Cells Using Nanoliter Droplets. Cell. 2015 May 21;161(5):1202-1214).
The manuscript provides an excellent overview of the contribution of scRNA-Seq to the field and would be a valuable contribution to the community, subject to addressing the following concerns and minor suggestions.
- The number of studies that have been published across the broad spectrum of eye diseases is remarkable and Table 1 provides a helpful compilation of vision related scRNA-Seq studies. However, although this is an expansive review it is inappropriate to suggest that it provides ‘a comprehensive review of scRNA-seq application advancement in ocular physiology and pathology’ because many pertinent studies have been omitted, eg:
Inge Van Hove et al. Single-cell transcriptome analysis of the Akimba mouse retina reveals cell-type-specific insights into the pathobiology of diabetic retinopathy 10.1007/s00125-020-05218-0
Wang SK et al. Single-cell multiome of the human retina and deep learning nominate causal variants in complex eye diseases. Cell Genom. 2022 Aug 10;2(8):100164. 10.1016/j.xgen.2022.100164
Indeed, many of the studies mentioned in the text are not included in Table 1.
It is understandable that some publications are not included because, as the authors highlight, there are so many studies. However, it should be decided whether this is a truly comprehensive review (and include all studies in Table 1), or state the criteria for inclusion of studies in the review. Related to this point, clarify the meaning of ‘typical’ in the description of Table 1 – does this mean exemplar studies in each area?
- It would be helpful to mention the publicly accessible datasets and the possibility for researchers to interrogate them to address their own research eg EBI Single Cell Expression Atlas, CZ cellxgene, The Human Cell Atlas, Tabula sapiens and Tabula muris etc.
Minor suggestions/typos:
Comment on single nucleus versus single cell and the use of different technology platforms.
Line 34, 36: Suggest avoiding the term ‘cluster’ in for example ‘corneal cell cluster’ to avoid confusion with the description of ‘clusters’ of cells in UMAP visualisations.
Line 56 – delete ‘deeply’
Line 59: Rephrase ‘Burgeoning literature employing scRNA-seq technology in ocular research has been reported till now, in this review…,’
Line 73: Reword to more accurately explain that RNA from each cell is identified by incorporation of a unique cell barcode during reverse transcription. (The RNA is not captured following barcoding)
‘Each cell is then tagged with a unique barcode during cDNA synthesis, enabling researchers to track and differentiate RNA molecules from each cell. Following barcoding, RNA is captured, reverse transcribed into cDNA, and amplified to increase sequencing sensitivity.’
Line 79. Need to mention differential expression analysis
Line 146. Clarify that identification of the role of ATOH7 in retinal ganglion development pre-dated scRNA-Seq (see eg Brown NL et al Development. 2001 Jul;128(13):2497-508. doi: 10.1242/dev.128.13.2497), but that scRNA-seq has added to our knowledge.
Line 183 – Split up overly long sentence
Line 326 use of ‘we’ inappropriate – it was other authors who speculated
Line 344 alternations should be alterations
Line 443 ‘progressive dysfunction of retinal microvascular’ change to ‘progressive dysfunction of retinal microvasculature’
Fig 1. Data analysis and visualisation appears to have electropherogram – use a more appropriate image!
Any papers recommended in the report are for reference
only. They are not mandatory. You may cite and reference other papers
related to this topic.
Although the English is generally of high standard, it requires some minor improvements. For example, check the use of tenses, eg in Figure legends
Reviewer 3 Report
Comments and Suggestions for Authors
This manuscript presents a comprehensive and in-depth overview of recent developments in the application of single-cell RNA sequencing (scRNA-seq) within ocular physiology and pathology. It emphasizes the technology’s utility in elucidating cellular heterogeneity, identifying novel biomarkers, and uncovering molecular mechanisms underlying ocular diseases such as glaucoma, myopia, uveitis, retinal disorders, and ocular neoplasms.
Strengths
-
The manuscript offers a thorough and current synthesis of scRNA-seq applications, establishing its significant relevance to ophthalmologic research.
-
The authors adeptly articulate the translational potential of scRNA-seq, particularly in the realms of therapeutic target identification and biomarker discovery.
-
The integration of disease-specific case studies and associated gene/pathway analyses adds considerable depth and rigor to the manuscript.
-
The use of well-labeled schematic figures enhances the reader’s comprehension of complex methodological processes and findings.
Weaknesses
-
While the manuscript is commendably detailed, the volume of information may, at times, overwhelm the reader. A more concise synthesis and clearer structural segmentation could significantly improve the overall readability.
-
Certain sections, especially those pertaining to individual ocular conditions (e.g., glaucoma, AMD, uveitis), would benefit from a more critical evaluation of methodological limitations and a comparative analysis with other omics-based approaches.
-
Although the text briefly addresses batch effects and spatial context limitations, it does not sufficiently explore emerging methodologies or mitigation strategies to address these challenges.
-
Minor grammatical and linguistic refinements would enhance the manuscript’s clarity and elevate its professional tone.
Minor Comments:
-
Consider restructuring the disease-focused sections into subcategories explicitly delineated by technological advancements, diagnostic applications, and therapeutic implications.
-
Ensure that abbreviations are defined upon first use and applied consistently throughout the manuscript, as inconsistencies were observed.
-
Suggested corrections:
-
“the CUCDC66 was consistently expressed” should read “CCDC66 was consistently expressed.”
-
The sentence “Recent advancements in scRNA-seq have provided novel insights into...” might be more effectively rendered as “Recent advancements in scRNA-seq offer novel insights into...”
-
-
Some references (e.g., Figure 3 and mentions of BioRender) could be more seamlessly integrated into the narrative by providing brief contextual explanations within the main text.
Recommendation
This manuscript constitutes a valuable and timely contribution to the field of ophthalmic research, particularly by underscoring the transformative potential of single-cell RNA sequencing. In light of its scholarly merit and comprehensive scope, I recommend minor revisions aimed at enhancing clarity, incorporating critical methodological perspectives, and refining the manuscript’s linguistic presentation prior to publication.
Comments on the Quality of English LanguageThe English could be improved in some sentences.
Round 2
Reviewer 1 Report
Comments and Suggestions for Authors
The revision is not satisfying. The authors did not focus the problems arised from the reviewer and most of the criticisms still persist.
Comments on the Quality of English LanguageThe english should be improved.
